

# The effect of postoperative oral antibiotic therapy on the incidence of postoperative endophthalmitis after phacoemulsification surgery in dogs: 368 eyes (1997–2010)

Meg D. Sorhus[1], Amanda Corr[1], Xiaocun Sun[2] and Daniel A. Ward[1]

[1] Department of Small Animal Clinical Sciences, University of Tennessee College of Veterinary Medicine, Knoxville, Tennessee, United States of America
[2] Office of Information Technology, University of Tennessee, Knoxville, TN, USA

## ABSTRACT

**Purpose:** To assess the effectiveness of postoperative administration of oral antibiotics at reducing the incidence of endophthalmitis following phacoemulsification cataract extraction in dogs.

**Methods:** Medical records of the University of Tennessee College of Veterinary Medicine were reviewed for cases having undergone phacoemulsification and divided according to whether or not they had received oral antibiotics postoperatively. Records were then evaluated for a diagnosis of endophthalmitis and incidence rates between the group receiving postoperative oral antibiotics and the group not receiving postoperative oral antibiotics were compared.

**Results:** A total of 215 patients (368 eyes) were identified by the search. One-hundred twelve patients (197 eyes) were treated with oral antibiotics postoperatively. One-hundred and three patients (171 eyes) were not treated with oral antibiotics postoperatively. Three cases of endophthalmitis were identified, with one in the antibiotic-treated group and two in the non-antibiotic treated group ($P > 0.05$, Fisher's exact test).

**Conclusions:** The overall incidence of endophthalmitis at the University of Tennessee from 1997–2010 was 0.82%. The rate of post-phacoemulsification endophthalmitis was unaffected by the postoperative administration of oral antibiotics.

Corresponding author
Daniel A. Ward, dward@utk.edu

## INTRODUCTION

Infectious endophthalmitis is one of the most devastating complications of phacoemulsification cataract extraction in both human and veterinary ophthalmology. In humans, post-phacoemulsification endophthalmitis results in visual acuity of 20/200 or worse in 15–30% of cases, and 10% are left with no useful vision (20/800 or less) (*Behndig et al., 2013*; *Durand, 2013*). The rate of post-cataract extraction endophthalmitis ranges from 0.012% to 0.56% in the human literature (*Liesegang, 2001*; *Ciulla, Starr & Masket, 2002*; *Kamalarajah et al., 2004*; *Li et al., 2004*; *Taban et al., 2005*; *Wejde et al., 2005*;

*Ou & Ta, 2006*; *Rosha et al., 2006*; *Cao et al., 2013*; *Rudnisky, Wan & Weis, 2014*), with two large meta-analyses reporting average rates of 0.134% (*n* = 6,686,169; *Cao et al., 2013*) and 0.128% (*n* = 3,140,560; *Taban et al., 2005*). Rates from 0–1.4% have been published in dogs (*Sigle & Nasisse, 2006*; *Johnstone & Ward, 2005*; *Azoulay et al., 2013*; *Ledbetter, Spertus & Kurtzman, 2018*; *Lacerda et al., 2018*). The presumed sources of ocular infection in humans are the eyelids and conjunctival surface (*Speaker & Menikoff, 1991*) and risk factors include advanced age, immunosuppressive comorbidities (*e.g.*, diabetes mellitus), and intraoperative complications (*Sengillo et al., 2020*); the same are presumed to be true for veterinary patients (*Ledbetter, Spertus & Kurtzman, 2018*). Although variations do exist, standards of care for prevention of infection during cataract surgery among human medical institutions have been suggested (*Rosha et al., 2006*; *Rudnisky, Wan & Weis, 2014*; *Behndig et al., 2013*). Many veterinary practices likely have been extrapolated from this data. However, there is still much opportunity to investigate the best practices for prevention of endophthalmitis in veterinary patients undergoing cataract surgery.

While intracameral antibiotic administration has found widespread acceptance as bacterial endophthalmitis prophylaxis (*Haripriya, Baam & Chang, 2017*), the most widely cited strategy, and the approach with the highest level of evidence of efficacy, is the preoperative use of 5% povidone-iodine on the periocular and ocular surfaces (*Speaker & Menikoff, 1991*; *Taylor et al., 1995*; *Liesegang, 2001*; *Ciulla, Starr & Masket, 2002*; *Mayer et al., 2003*; *Ou & Ta, 2006*; *Rosha et al., 2006*; *Sengillo et al., 2020*). Perioperative administration of topical antibiotics is common in both human and veterinary cataract surgery, despite a lack of evidence that they reduce the incidence of post-operative bacterial endophthalmitis (*Grzybowski et al., 2016*; *ESCRS Endophthalmitis Study Group, European Society of Cataract & Refractive Surgeons, 2007*; *Huang et al., 2016*).

Although there is no evidence that postoperative systemic antibiotic administration reduces the incidence of postoperative bacterial endophthalmitis, many veterinary ophthalmologists list postoperative systemic antibiotic administration as part of their perioperative protocols (*Paulsen et al., 1986*; *Boldy, 1988*; *Taylor et al., 1995*; *Ledbetter, Millichamp & Dziezyc, 2004*; *Sigle & Nasisse, 2006*; *Hazra et al., 2008*; *Gift et al., 2009*; *Ledbetter, Spertus & Kurtzman, 2018*). Postoperative oral antibiotics are not used routinely following cataract extraction in humans; in one study of endophthalmitis prophylaxis practices, postoperative oral antibiotics were not used in any of the 75,318 cataract extractions that were reviewed (*Rudnisky, Wan & Weis, 2014*). Prophylactic use of postoperative systemic antibiotics is particularly problematic due to the potential for antibiotic resistance, adverse side effects, additional cost and additional medication for owners to administer. Up to 50–60% of operations in human general surgery are estimated to be associated with over-use, under-use or misuse of antibiotics (*Bratzler et al., 2005*; *De Almeida et al., 2018*). Recent guidelines made by the *American Journal of Veterinary Internal Medicine* have been made in response to recognition of the importance of veterinarians' role in reducing antimicrobial resistance whenever possible (*Weese et al., 2015*).

The purpose of this retrospective study was to determine whether prolonged postoperative oral antibiotic administration is effective in reducing the incidence of

endophthalmitis after phacoemulsification in dogs, and we hypothesized that a statistically significant effect would not be found.

## MATERIALS AND METHODS

Medical records of patients that had phacoemulsification performed at the University of Tennessee Veterinary Teaching Hospital (UTVTH) between 1/1/1997 and 12/31/2010 were reviewed for procedure codes of "cataract surgery" or "phacoemulsification" and a diagnosis code of "endophthalmitis." This period of time was evaluated because in that timeframe some of the cataract patients at UTVTH received oral antibiotics following cataract extraction while others did not. Cases receiving pre- and/or postoperative topical antibiotics other than neomycin-polymixin B-gramicidin, cases with evidence of postoperative corneal ulceration and patients that did not return for at least three postoperative follow-up appointments were excluded from the study. Records were evaluated for breed, patient age, type of intraocular lens (IOL) that was implanted, level of experience of surgeon (*i.e.*, diplomate of the American College of Veterinary Ophthalmologists (ACVO) *vs.* ophthalmology resident), diabetic status, postoperative ocular hypertension (defined as intraocular pressure (IOP) > 25 mmHg within 24 h of surgery) and operative time.

Initial ocular examinations consisted of slit-lamp biomicroscopy, indirect ophthalmoscopy, Schirmer tear testing, fluorescein staining, applanation tonometry, 15 MHz ocular ultrasonography, and flash electroretinography (ERG). Unless otherwise specified, ultrasonography and electroretinography were not performed on recheck examinations. Urine culture was performed on all diabetic patients, and if urinary tract infections were identified they were treated prior to surgery. Examinations were performed by an ACVO diplomate, a resident in an ACVO-approved residency program, or both. Surgeries were performed by an ACVO diplomate, a third year veterinary ophthalmology resident, or a first or second year veterinary ophthalmology resident under the direct supervision of an ACVO diplomate. A total of eight different surgeons performed surgeries included in the timeframe of this study (two ACVO dipl., six residents). Surgery was standard one- or two-handed phacoemulsification, per surgeon's preference, and instruments were not changed between eyes of bilateral surgeries. In general, pre-operative protocols consisted of treatment with topical anti-inflammatory medication (steroidal and/or nonsteroidal) BID for approximately 7 days prior to surgery. With slight variation in timing, treatment protocols for the day prior to and morning of surgery consisted of topical 1% prednisolone acetate QID, topical 0.03% flurbiprofen QID, topical 0.175% neomycin-0.0025% gramicidin-10,000 units/mL polymixin B QID, topical 2.5% phenylephrine QID, and systemic flunixin meglumine (0.44 mg/kg IV immediately prior to anesthetic induction) or carprofen (2.2 mg/kg SQ 2 h prior to anesthetic induction). Cefazolin (22 mg/kg IV) was administered pre- and intra-operatively beginning at anesthetic induction and continued every 90 min until the end of surgery. Polymethylmethacrylate (PMMA) or foldable acrylic intraocular lenses (IOLs) were

inserted within the lens capsule when appropriate at the discretion of the surgeon. The types of PMMA and acrylic IOLs used did not change over the study period. Postoperative protocols generally consisted of topical anti-inflammatories, antibiotics, parasympatholytics, and artificial tears. In addition, some patients received subconjunctival corticosteroids (0.5–1.0 mg dexamethasone sodium phosphate) or systemic corticosteroids (prednisolone 0.5 mg/kg PO QD-BID for 5–10 days). No specific reasoning for opting to administer subconjunctival corticosteroids was discernable from the medical records. Systemic corticosteroids were given in cases of posterior capsule disruption or severe uveitis the day following surgery.

Oral postoperative antibiotics were used in some cases but not in others based upon surgeon's preference and training; review of the medical records did not identify any patient, surgical, environmental, client, or other variables that might have influenced this decision. When prescribed, oral antibiotics were usually either amoxicillin ($n = 34$), cefadroxil ($n = 62$) or cephalexin ($n = 13$) dosed at 20–30 mg/kg PO BID for 7–10 days, based upon surgeon's preference, availability at our pharmacy, and cost. Three dogs were on different antibiotics at surgical admission for non-ocular reasons. One was on cefpodoxime 11 mg/kg PO QD for 30 days to treat pyoderma, one was on enrofloxacin five mg/kg PO BID for seven days to treat a prostatic abscess, and one was on amoxicillin/clavulanic acid 13 mg/kg PO BID for 14 days. The oral antibiotic therapy for these dogs was not changed following cataract surgery.

All operated patients were rechecked 1 day postoperatively. Additional routine recheck examinations were planned for approximately 7 days, 21 days, and 56 days postoperatively; actual ranges of these rechecks were 4–11 days, 20–29 days, and 36–78 days respectively for the antibiotic-treated group, and 5–10 days, 19–27 days, and 40–57 days for the non-antibiotic-treated group. Additional recheck exams were conducted if deemed necessary by the attending clinician or requested by the owner. Patients that did not return to UTCVM for at least the first three scheduled rechecks were not included in the study. Postoperative endophthalmitis was diagnosed by an ACVO diplomate based upon clinical examination finding of severe anterior uveitis (characterized by 3+/4+ or 4+/4+ anterior chamber flare and cells) with hypopyon, decreased vision (as indicated by loss of menace response), and ocular pain beyond what would be expected postoperatively, with or without a positive aqueous humor culture, as per *Rudnisky, Wan & Weis (2014)*. Histopathology of one enucleated eye was performed by a diplomate of the American College of Veterinary Pathologists (ACVP) or a resident in an ACVP-approved residency.

Each eye that underwent phacoemulsification was considered the experimental unit for the purpose of statistical analysis. Descriptive statistics were reported as median together with either range or 25th and 75th percentiles. Chi square (or 2-tailed Fisher's exact test when contingency table values <5) was used to compare categorical data (including the rate of endophthalmitis in dogs that did receive postoperative systemic antibiotics with those that did not) using commercially available software[1].
The Mann-Whitney U test was used to compare continuous data between the two groups. $P < 0.05$ was considered statistically significant.

[1] SigmaPlot 14, Systat Software, San Jose, CA, USA.

**Table 1 Comparison of group receiving postoperative oral antibiotics with group not receiving postoperative oral antibiotics.**

|  | Group receiving postoperative oral antibiotics | Group NOT receiving postoperative oral antibiotics | P |
|---|---|---|---|
| No. of patients | 112 | 103 |  |
| No. of eyes | 197 | 171 |  |
| No. of eyes diagnosed with endophthalmitis | 1 | 2 | 0.599 |
| Breeds[1] |  |  |  |
|   Mixed breed dog | 20 (18%) | 16 (16%) |  |
|   Boston terrier | 14 (13%) | 15 (15%) |  |
|   Cocker spaniel | 14 (13%) | 8 (8%) |  |
|   Miniature poodle | 11 (10%) | 6 (6%) |  |
|   Bichon frise | 7 (6%) | 6 (6%) |  |
|   Miniature schnauzer | 7 (6%) | 6 (6%) |  |
|   Toy poodle | 7 (6%) | 7 (7%) |  |
| Age (median (25%, 75%); yrs) | 7 (4, 9) | 7 (4,10) | 0.73 |
| No. eyes with IOLs |  |  |  |
|   PMMA (n (% of eyes w/known IOL status)) | 109 (64%) | 117 (68%) |  |
|   Acrylic | 18 (11%) | 26 (15%) | 0.09 |
|   Aphakic | 43 (25%) | 28 (16%) |  |
|   Not recorded | 27 | 0 |  |
| No. eyes operated by faculty/residents | 92/105 (47%/53%) | 83/88 (49%/51%) | 0.26 |
| No. diabetic patients | 18 (16%) | 23 (22%) | 0.18 |
| No. eyes with IOP > 25 mmHg within 24 h of surgery | 46 (23%) | 35 (20%) | 0.27 |
| Operative time/eye (median (25%, 75%); mins) | 62 (51, 70) | 63 (50,75) | 0.98 |
| Phacoemulsification machine[2] | AMO diplomax (n = 197) | AMO diplomax (n = 152) Alcon infiniti (n = 19) |  |

**Notes:**
[1] Twenty non-listed breeds in the antibiotic-treated group and 22 non-listed breeds in the non-antibiotic-treated group had fewer than six individuals each.
[2] Both phacoemulsification machines used in this study utilize peristaltic pump technology. All endophthalmitis cases were operated using the AMO Diplomax unit.

## RESULTS

A total of 215 patients and 368 eyes were identified by the medical records search as satisfying inclusion and exclusion criteria. Median follow-up times were 412.5 days in the antibiotic-treated group (range 48–1,800 days) and 220 days (range 24–696 days) in the non-antibiotic treated group. The distribution of breeds and IOL types were similar between the two groups (see Table 1). Similarly, there were no differences between the two groups with respect to age, proportion of eyes operated by ACVO diplomates vs. residents, proportion of diabetic patients, proportion of eyes with IOP > 25 mmHg within 24 h of surgery, or operative time (see Table 1).

One-hundred and twelve patients were treated and 103 patients were not treated with systemic oral antibiotics post-operatively. Among the 112 antibiotic-treated patients, 85 were operated bilaterally and 27 were operated unilaterally for a total of 197 eyes in the antibiotic-treated group. Among the 103 patients not treated with postoperative oral antibiotics, 68 were operated bilaterally and 35 were operated unilaterally for a total of 171 eyes in this group. During the study period, three eyes of three different patients (one eye

from the antibiotic-treated group and two from the non-antibiotic-treated group) developed postoperative endophthalmitis, for an overall incidence of 3/368, or 0.82%. There was no statistically significantly difference in incidence rates between the two groups ($P = 0.599$; power at $\alpha = 0.05$: 0.370 to detect a five-fold difference, 0.763 to detect a ten-fold difference).

## ENDOPHTHALMITIS CASES

The first affected patient (from the antibiotic-treated group) was a 12 year old nondiabetic female spayed mixed breed dog with bilateral immature cataracts. Apart from the cataracts, ocular examination was within normal limits. Preoperative ERG was normal with a-wave: b-wave amplitude > 100 µV in both eyes (OU), and ocular ultrasonography showed mature cataracts with no evidence of retinal detachment, lens capsule rupture or vitreal degeneration bilaterally. The dog underwent uneventful bilateral phacoemulsification. This patient was left aphakic OU; the reason for that decision was not elucidated in the medical record, but the surgery report did not indicate lens instability nor posterior capsular tears. The surgical time in this patient was 40 min in each eye. Immediate postoperative treatment consisted of a single subconjunctival injection of 0.4 mg dexamethasone sodium phosphate, prednisolone acetate (one drop OU q 4 h), neomycin-polymixin B-gramicidin (one drop OU q 8 h), prednisone (one mg/kg PO q 12 h) and cephalexin (25 mg/kg PO q 8 h). On postoperative day (POD) 1 an incisional wound leak was present and cyanoacrylate tissue adhesive was applied. The incision was still leaking on POD 2, so the patient was taken back to surgery and the sutures were removed and replaced without complication. The dog was discharged on POD 3 on a regimen of prednisolone acetate (one drop OU q 6 h), neomycin-polymixin B-gramicidin (one drop OU q 8 h), tropicamide (one drop OU q 12 h), oral prednisone at a tapering dose (one mg/kg PO q 12 h × 4 days, then q 24 h), and cephalexin (25 mg/kg PO q 8 h for 10 days). On POD 15, recheck examination demonstrated that there was no aqueous flare OU, no wound leaks OU, no corneal ulcers OU, and IOPs of 26 mmHg in the right eye (OD) and 23 mmHg in the left eye (OS). Vision was considered good based on positive menace responses OU and ability to successfully navigate an obstacle course in the exam room. Prednisolone acetate was decreased to one drop OU q 8 h, neomycin-polymixin B-gramicidin was continued at one drop OU q 8 h, and tropicamide was decreased to q 24 h. Oral prednisone was decreased to one mg/kg PO q 48 h for another week and then discontinued.

On POD 19, the patient returned after the owner noticed acute onset redness, cloudiness, pain, and periocular swelling OS. Exam OD showed no aqueous flare, no wound leaks, no corneal ulcers, and an IOP of 20 mmHg. The OS had conjunctival and episcleral injection, aqueous leakage at the incision site, 3+ diffuse corneal edema, 4+ aqueous flare and cells, hypopyon, fibrin in the anterior chamber and miosis OS. The dog was judged to be blind OS based on a negative menace response. The IOP was 30 mmHg OS. The presumptive diagnosis was endophthalmitis and secondary glaucoma. Aerobic, anaerobic and fungal cultures were obtained from the wound leak and from an aqueous humor paracentesis. Both sites yielded positive growth of *Enterococcus faecalis*

that was sensitive to all antibiotics tested (ampicillin, chloramphenicol, penicillin, vancomycin, gentamicin and streptomycin); anaerobic and fungal cultures were negative. An ultrasound OS identified a complete retinal detachment with cellular infiltrate in the vitreal chamber. Due to the ocular pain and poor prognosis for vision, the left eye was enucleated. The eye was submitted for histopathologic examination, which revealed corneal edema, large numbers of neutrophils admixed with fibrin in the anterior and posterior chambers, neutrophilic infiltration of the iris, ciliary body and choroid, retinal detachment, degeneration of the photoreceptor layer, and retinal pigment epithelial hypertrophy. No microorganisms were identified on histopathology.

The second affected patient (from the non-antibiotic-treated group) was a 12 year old diabetic male castrated mixed breed dog with mature cataracts OU. In addition to the cataracts, lens instability due to zonular disruption was noted OD. Therefore, an intracapsular extraction was performed OD and an IOL was not placed. Surgery OS was uneventful and a PMMA IOL was placed in that eye. Operative times were 45 mins OD, 85 mins OS. Post-operatively, the patient was started on prednisolone acetate (one drop OU q 4 h), neomycin-polymixin B-gramicidin (one drop OU q 4 h), and ocular lubricating gel (1/4″ OU q 4 h). The dog was discharged on POD 2 on a regimen of prednisolone acetate (one drop OU q 6 h), neomycin-polymixin B-gramicidin (one drop OU q 6 h), tropicamide (one drop OU q 8 h), and ocular lubricating gel (1/4″ OU q 4 h). Systemic antibiotics were not prescribed. Recheck exams on PODs 9 and 23 were normal (no flare OU, normal IOPs OU, no ulcers OU) with the exception of retinal hemorrhage OS, which was attributed to diabetic retinopathy (*Landry, Herring & Panciera, 2004*). The tropicamide and neomycin-polymixin B-gramicidin were discontinued and the prednisolone acetate was decreased to q 24 h.

On POD 30, the patient was re-presented for a scheduled recheck and had 3+ aqueous flare and cells and hypopyon OS. The IOP was five mmHg. The patient was diagnosed with anterior uveitis and presumptive endophthalmitis OS. The patient was discharged on enrofloxacin (five mg/kg PO BID), prednisolone acetate (one drop up to QID OS) and atropine (one drop OS TID). On POD 35 flare and hypopyon were reduced, though fibrin and hyphema were present. The IOP OS was too low to read. All medications were continued. The endophthalmitis OS continued to show subjective improvement and on POD 44 the enrofloxacin was discontinued while the prednisolone acetate and atropine were continued as previously prescribed. By POD 65 the endophthalmitis was considered resolved with no hyphema, fibrin, flare or hypopyon present. The prednisolone acetate was decreased (TID × 2 weeks and then BID × 2 weeks) and the atropine was decreased (BID). All medications were discontinued on POD 86, at which time menace responses were positive OU and functional vision was deemed to be excellent. The patient was lost to follow-up after this visit.

The third affected patient (also from the non-antibiotic-treated group) was an 11 year old nondiabetic male castrated toy poodle with a late immature cataracts OD and hypermature cataract OS. Medical history included recurrent pyodermas, but the dog was otherwise in good health. Intraoperatively significant zonular disruption was noted OS, so following lens removal the lens capsule was also removed and no IOL was placed in

that eye. Surgery OD was uneventful and a PMMA IOL was placed in that eye. Operative times were 60 mins OS and 65 mins OD. Postoperatively, the patient was started on prednisolone acetate (one drop OU q 4 h), neomycin-polymixin B-gramicidin (one drop OU q 4 h), and ocular lubricating gel (1/4″ OU q 4 h). Operative times were 45 mins OD and 85 mins OS. The evening of surgery the IOP OS rose to 26 mmHg, so the dog was placed on dorzolamide/timolol one drop OS BID. On POD 1 the IOP had dropped to 9 mmHg OS, and the dog was discharged on a regimen of prednisolone acetate (one drop OU q 6 h), neomycin-polymixin B-gramicidin (one drop OU q 6 h), tropicamide (one drop OU q 8 h), ocular lubricating gel (1/4″ OU q 4 h) and dorzolamide/timolol (one drop OS q 12 h). Systemic antibiotics were not prescribed. Recheck examinations on PODs 8, 21, 44 and 56 were unremarkable apart from a mild IOP increase OS on POD 21 (25 mmHg). By POD 44 the IOP OS had returned to normal at six mmHg. Between PODs 8 and 56 all medications were gradually weaned and discontinued.

On POD 87 the dog was presented for acute onset blepharospasm and discharge OS of 2–3 days' duration. Ocular exam revealed hypopyon completely filling the anterior chamber so that the iris was only visible at its periphery. The eye was diagnosed with endophthalmitis and treated with topical ciprofloxacin (one drop OS q 2 h), and oral enrofloxacin (five mg/kg PO BID) and amoxicillin/clavulanic acid (13 mg/kg PO BID). The dog was rechecked on POD 89, and no improvement was noted. An ocular ultrasound revealed hyperechoic material consistent with purulence and/or blood in the anterior chamber and vitreous cavity, with no obvious retinal detachment. We recommended an anterior chamber tap for cytology and culture, but the owner declined and opted to have the eye enucleated at the referring veterinarian's office. We requested that the referring veterinarian obtain an intraocular culture at the time of enucleation and to have the eye returned to us for histopathology, but neither of these requests were honored.

## DISCUSSION

Our data indicate that endophthalmitis following phacoemulsification cataract extraction is a rare event in dogs, occurring in three out of 368 at risk eyes in this study (0.82%). This is in general agreement with previous studies in dogs (*Johnstone & Ward, 2005*; *Sigle & Nasisse, 2006*; *Azoulay et al., 2013*; *Ledbetter, Spertus & Kurtzman, 2018*) but higher than most studies in humans (*Liesegang, 2001*; *Ciulla, Starr & Masket, 2002*; *Kamalarajah et al., 2004*; *Li et al., 2004*; *Taban et al., 2005*; *Wejde et al., 2005*; *Ou & Ta, 2006*; *Rosha et al., 2006*; *Cao et al., 2013*; *Rudnisky, Wan & Weis, 2014*). Furthermore, our data did not reveal that postoperative administration of systemic antibiotics protected against endophthalmitis. With the diagnosis of endophthalmitis being so rare in both groups, one must be wary of statistical serendipity hiding significant differences. The power to detect treatment effects of five-fold and ten-fold were 0.360 and 0.763, respectively. Our findings would be more convincing had the power exceeded 0.80, which would have required an increase in sample size. However, it can be calculated that an additional 6 endophthalmitis cases would have had to occur in the group not receiving systemic antibiotics (with no additional cases in the group that did receive systemic antibiotics) to demonstrate a statistically significant protective effect of systemic antibiotics. Given that

we only found two cases in the untreated cohort of 171 eyes, it is highly unlikely that our conclusions would change if the study were extrapolated to thousands of cases.

The cases in this report developed more than a week postoperatively, analogous to what some authors in the human literature designate "chronic post-phacoemulsification endophthalmitis" (*Durand, 2013*); other authors describe cases occurring within 6 weeks of surgery as "acute" and those occurring more than 6 weeks after surgery as "delayed onset" (*Miller et al., 2005*). Most cases of endophthalmitis following phacoemulsification probably occur following introduction of infectious agents from the conjunctiva and eyelid margins (*Sherwood et al., 1989*; *Tervo et al., 1999*; *Liesegang, 2001*; *Ledbetter, Millichamp & Dziezyc, 2004*), although Lacerda et al demonstrated that sources of contamination other than the ocular surface do occur in canine phacoemulsification (*Lacerda et al., 2018*). The third patient affected with endophthalmitis in this report deserves particular attention. The diagnosis in that case was made on POD 87, which is an extraordinarily long time even for chronic or delayed-onset post-phacoemulsification endophthalmitis (*Durand, 2013*), and we question whether that case truly represents an infection related to surgery. We included that case simply because it did fit our predetermined inclusion criteria; if one were to eliminate that case it would only strengthen our conclusion that postoperative antibiotics do not reduce the incidence of postoperative endophthalmitis as that case was in the group that did not receive antibiotics after surgery. If that case were eliminated, the total endophthalmitis incidence in our study would be 2/366 (0.55%), with 1/197 (0.51%) in the antibiotic-treated group and 1/169 (0.59%) in the non-antibiotic-treated group.

In humans the most common offending organisms are coagulase negative *Staphylococci* in acute cases (*i.e.*, POD 2–7) and *Propionibacterium acnes* in more chronic cases (*Durand, 2013*). The single culture-proven case in the present study was an *Enterococcus*, but the number of reported cases of post-phacoemulsification endophthalmitis in dogs is too limited to make any conclusions about predominance of any specific organism (*Johnstone & Ward, 2005*; *Sigle & Nasisse, 2006*; *Ledbetter, Spertus & Kurtzman, 2018*). It has been shown that introduction of organisms into the eye during phacoemulsification is rather common in dogs and humans, yet development of endophthalmitis is rare (*Dickey, Thompson & Jay, 1991*; *Taylor et al., 1995*; *Tervo et al., 1999*; *Ledbetter, Millichamp & Dziezyc, 2004*; *Ledbetter, Spertus & Kurtzman, 2018*). This indicates that the eye has effective mechanisms for clearing organisms from the anterior chamber prior to the development of infection, with the most important mechanism being constant turnover of aqueous humor (*Durand, 2013*).

Reported risk factors for developing post-phacoemulsification endophthalmitis have been identified in humans and include posterior capsule disruption, incisional wound leak, concurrent diabetes mellitus, concurrent periocular disease, immunoincompetence, and advanced age (*Aaberg et al., 1998*; *Liesegang, 1999*; *Liesegang, 2001*; *Li et al., 2004*; *Wejde et al., 2005*; *Ou & Ta, 2006*; *Cao et al., 2013*; *Rahmani & Eliott, 2018*). It is interesting to note that all of our endophthalmitis cases each had one of these risk factors. Case 1 had a wound leak, case 2 was a diabetic, and case 3 had posterior capsule removal due to zonular instability, also necessitating anterior vitrectomy. Other factors

reported to influence postoperative endophthalmitis rates in humans include surgical technique (with manual extracapsular extraction associated with higher rates than phacoemulsification), corneal incision type (with clear corneal incisions associated with higher rates than scleral tunnel incisions), and choice of intraocular lens (with injectable lenses associated with lower rates than non-injectable lenses) (*Mayer et al., 2003*; *Miller et al., 2005*; *Ou & Ta, 2006*; *Rosha et al., 2006*; *ESCRS Endophthalmitis Study Group, European Society of Cataract & Refractive Surgeons, 2007*).

Among methods of reducing the incidence of post-phacoemulsification endophthalmitis, surgical preparation with 5% povidone-iodine is most consistently supported in the literature (*Speaker & Menikoff, 1991*; *Liesegang, 1999*; *Schmitz et al., 1999*; *Liesegang, 2001*; *Ciulla, Starr & Masket, 2002*; *Ang & Barras, 2006*; *Ou & Ta, 2006*). There is no evidence that prolonged postoperative systemic antibiotics are beneficial in reducing the incidence of postoperative endophthalmitis in humans (*Liesegang, 2001*; *Ang & Barras, 2006*), and our data suggest the same is true in dogs. The numerous untoward effects of injudicious use of antibiotics makes it clear that without evidence of active infection systemic antibiotics should not be administered after phacoemulsification in dogs. For preoperative antimicrobial prophylaxis in general surgery, the Surgical Infection Prevention Guideline Writers Workgroup advocates administering a first intravenous antimicrobial dose within 60 min before surgical incision and discontinuing antimicrobials no more than 24 h after the end of surgery (*Bratzler & Houck, 2005*). It seems prudent to follow these recommendations for phacoemulsification in dogs.

Preoperative topical antibiotics are also widely used as endophthalmitis prophylaxis, and while reports of efficacy are conflicting most cataract surgeons continue to use them (*Lehmann et al., 1997*; *Liesegang, 1999*; *Ta et al., 2002*; *de Kaspar et al., 2004*; *Ou & Ta, 2006*; *Huang et al., 2016*). In one review of perioperative practices aimed at preventing postoperative endophthalmitis, 76.5% of cases received preoperative topical antibiotics, but the specific agents utilized were not specified (*Inoue et al., 2018*). Another meta-analysis reported wide variability in specific agents used, including fluoroquinolones, aminoglycosides, cephalosporins, and chloramphenicol (*Haripriya, Baam & Chang, 2017*). Some studies have also shown a protective effect of subconjunctival antibiotics and intracameral antibiotics (*Schmitz et al., 1999*; *Rosha et al., 2006*; *ESCRS Endophthalmitis Study Group, European Society of Cataract & Refractive Surgeons, 2007*), while other studies have found no benefit from these measures (*Ciulla, Starr & Masket, 2002*). The preponderance of the evidence from the most recent evaluations of intracameral antibiotic use following phacoemulsification in humans strongly suggest that they do have a protective effect against endophthalmitis, with the major limitations being availability of commercially available products approved for that use and complications associated with compounded products (*Haripriya & Chang, 2018*).

In some particular respects our data must be viewed with caution. Conclusions drawn in retrospective studies are only as reliable as the medical record entries that give rise to them; this variable cannot be controlled after the fact. As with all retrospectives, subjects could not be randomized into oral antibiotic-treated and non-antibiotic-treated groups. It would also have been optimal had all dogs in the antibiotic-treated group received the

same antibiotic. However, given that 109 of 112 dogs in this group received β-lactam antibiotics with similar spectra (as were two of the other three dogs who were already on oral antibiotics when admitted for surgery) it is unlikely that results would have been different had oral antibiotic administration been more consistent. Additionally, while 368 operated eyes would appear to be a relatively large sample size, it is suboptimal when studying a condition of extreme rarity; similar studies in the human literature usually include 10,000–100,000 subjects, and meta-analyses may include millions (*Aaberg et al., 1998*; *Mayer et al., 2003*; *Li et al., 2004*; *Wejde et al., 2005*; *Cao et al., 2013*). It would also be more convincing if all suspected cases had positive cultures, but it has been established in the human endophthalmitis literature that many cases proven to be associated with bacterial infection on PCR testing are negative on microbiological culture (*Therese, Anand & Madhavan, 1998*; *Kosacki et al., 2020*). Thus, bacterial endophthalmitis following phacoemulsification is often a clinical diagnosis based on ocular signs and time to occurrence (*Durand, 2013*; *Rudnisky, Wan & Weis, 2014*), and *Kamalarajah et al. (2004)* even refer to the syndrome as "presumed infectious endophthalmitis". Nevertheless, it should be considered the standard of practice that vitreal cultures and microbial PCR testing be performed on suspected postoperative endophthalmitis cases. It has been speculated that sterile endophthalmitis can occur following cataract extraction (Ledbetter et al., 2018), although most reviews of postoperative endophthalmitis presume it is due to bacterial infection, even if culture negative (*Kamalarajah et al., 2004*; *ESCRS Endophthalmitis Study Group, European Society of Cataract & Refractive Surgeons, 2007*; *Haripriya & Chang, 2018*). If postoperative endophthalmitis does occur in the true absence of bacterial infection, none of the proposed prophylactic strategies would be expected to be effective. Similarly, post-phacoemulsification endophthalmitis must be differentiated from toxic anterior segment syndrome (TASS). The two conditions have similar clinical signs, but TASS is marked by peracute onset (usually within 24 h of surgery), whereas post-phacoemulsification endophthalmitis can be delayed up to 6 weeks or more. Other distinguishing features include the degree of ocular discomfort, flare/cell/hypopyon response, and vision loss, all of which are minimal in TASS cases but severe in endophthalmitis cases (*Sengillo et al., 2020*).

## CONCLUSIONS

Our data support the hypothesis that prolonged postoperative antibiotic administration does not demonstrably reduce the rate of post-phacoemulsification endophthalmitis, and we recommend against this practice.

### Funding

The authors received no funding for this work.

## Competing Interests

The authors declare that they have no competing interests.

## Author Contributions

- Meg D. Sorhus performed the experiments, authored or reviewed drafts of the paper, and approved the final draft.
- Amanda Corr conceived and designed the experiments, performed the experiments, authored or reviewed drafts of the paper, and approved the final draft.
- Xiaocun Sun analyzed the data, authored or reviewed drafts of the paper, and approved the final draft.
- Daniel A. Ward conceived and designed the experiments, performed the experiments, analyzed the data, prepared figures and/or tables, authored or reviewed drafts of the paper, and approved the final draft.

## Data Availability

The raw data are available in the Supplemental File.

## Supplemental Information

Supplemental information for this article can be found online at http://dx.doi.org/10.7717/peerj.12305#supplemental-information.

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
