# Peer review of "The effect of postoperative oral antibiotic therapy on the incidence of postoperative endophthalmitis after phacoemulsification surgery in dogs: 368 eyes (1997–2010)"

_PeerJ, doi:10.7717/peerj.12305_

## Round 0.1 · original submission · Major Revisions

Two reviewers have evaluated this paper, and they include a number of important considerations. The authors must address all these points, and include the necessary changes.

Reviewer 1 ·

Basic reporting

The manuscript is well-written and clear.

Experimental design

This is an important subject matter. Clear and thoroughly described design.

Validity of the findings

Many of the authors conclusions are questionable as described in detail in the “General comments" section.

Limitation of this study are numerous and insufficiently acknowledged, including the lack of randomization, small sample size, and failure to confirm the diagnosis of endophthalmitis in 2 of 3 cases.

Additional comments

1. Lines 45 – 48: “The only strategy proven to decrease the rate of endophthalmitis is the preoperative use of 5% povidone-iodine on the periocular and ocular surfaces” is an oversimplification. For example, there are numerous publications describing statistically significant decreases in the rates of endophthalmitis with intracameral antibiotics given following phacoemulsification in humans (this is why it is considered the standard of care in many regions of the world).

2. Line 11-112: The lack of group randomization is a major limitation that is not fully acknowledged. A related critical issue is the statement “Oral post-operative antibiotics were used in some cases but not in others based upon surgeon’s preference” that is not well explained. What influenced this preference? Were there potentially patient, surgical, environmental, client, or other variables that might have influenced this decision in one direction or another, thus adding bias to the composition of your study groups?

3. Line 114: Similar to the statement above, what influenced surgeon preference for antibiotic selection? Since these are all lumped together in one group, what assurance can you provide a reader that each of these drugs would provide a similar degree of protection against endophthalmitis or at least a similar spectrum of action?

4. Is should be more clearly explained in the study that for most your cases you don’t know if these were bacterial, fungal, or sterile cases of endophthalmitis. It is also never clearly stated what you think you were trying to prevent with the antibiotic? Presumably, this would be bacterial endophthalmitis, but the lack of diagnostic confirmation in your endophthalmitis cases is problematic. While your discussion about PCR is understood and not disputed (although one can question the clinical significance of bacteria in some cases that are culture negative and PCR positive), you did neither for 66% of your endophthalmitis cases. If these were both sterile endophthalmitis cases, why would post-op antibiotics be expected to prevent them?

5. The single case with culture confirmation only included an aqueous sample which is inferior to vitreous for the diagnosis of post-op endophthalmitis and may actually have different organisms present. The other cases do not fit the “usual’ described pattern of infectious endophthalmitis in dogs, so to this review it seems like you are more accurately just comparing cases of unexplained hypopyon (vs true, confirmed endophthalmitis cases.)

6. While 0.82% might be in “general agreement” with the rate of endophthalmitis in some prior canine studies, it is notably (over 3x) higher than the largest study evaluating this occurrence in dogs (the Ledbetter et al. study from 2018) where they reported approximately 0.27% of eyes. Any thoughts on why this might be?

7. The study is under-powered and reports negative results. Thus, the statement that “postoperative administration of systemic antibiotics was not protective against endophthalmitis in our study as 1 case occurred in each the antibiotic-treated group and 2 in the non-antibiotic-treated group” and “systemic antibiotics should not be administered after phacoemulsification in dogs “ are over-reaching and, in this reviewer’s opinion, not supported by the data. While the reviewer may agree with these author conclusions, this study does not prove them.

8. Line 330: “while 329 operated eyes would appear to be a relatively large sample size”…I would ask to whom would this seem like a large sample size? This would appear much too small to any cataract surgeon if one is attempting to meaningfully answer the question the study was designed to address.

9. Line 337. You mention the limits of culture, but only cultured a single case (and not even the ideal ocular sample). Lack of diagnostic confirmation is a major study limitation. Culture at a minimum, and combined with PCR ideally, should be what is stated here as what would have improved this and future studies on the topic.

10. Can one conclusively (or at least convincingly) state that the dog that presented with endophthalmitis on post-op day 87 was a situation associated with the prior surgery? There are rather unusual situations in humans that can lead to post-op infection this far after surgery, but there are other potential explanation that having nothing to do with surgery as well. As you have no culture, PCR, or histologic confirmation, this uncertainty should be acknowledged.

Reviewer 2 ·

Basic reporting

Please see below 'General comments for the author'.

Experimental design

Please see below 'General comments for the author'.

Validity of the findings

Please see below 'General comments for the author'.

Additional comments

General comments:
It is a well written manuscript and an interesting topic that is important to discuss. You have a lot of cases over a 10 year period but unfortunately (or lucky enough) you only have three cases with septic endophthalmitis – and I’m not convinced that one of these cases had septic endophthalmitis (see below). Why exclude other topical antibiotics that NeoPolyBac(Gram) and would that include more septic endophthalmitis cases?
I would like more discussion on oral and topical antibiotic and what would you as a research team recommend in phaco cases. I would like to see your discussion have more information and discussion regarding antibiotics used in phaco dogs – the other veterinary endophthalmitis studies you are referring to – what topical antibiotics did they use? Topical NeoPolyGram will not penetrate the cornea and if you are reviewing newer phaco studies, a lot of institutions are using topical ciprofloxacin or ofloxacin these days due to studies showing that topical fluoroquinolones will penetrate the cornea and will be found in AC – and therefore treat potential septic endophthalmitis. I would like a discussion on what human phaco patients are being treated with regarding topical antibiotics, and what we are using in veterinary ophthalmology. A discussion regarding using oral antibiotic in diabetic cases versus non-diabetic would be recommended to add as well – see below for suggestion for urinalysis/culture results of your cases.

Specific comments/suggestions.
Abstract
1. 112 patients (197 eyes) were treated with oral antibiotics postoperatively. 103 patients (171 eyes) were not treated with oral antibiotics postoperatively.
Comment: You should not start a sentence with a number unless you write it: One-hundred twelve patients (197 eyes) …
2. Three cases of endophthalmitis were identified, with 1 in the antibiotic treated group and 2 in the non-antibiotic treated group (P>0.05, Fisher’s exact test).
Comment: All numbers under ten should be written: …, with one in the antibiotic treated group and two in the …

Introduction
Line 31: Instead of saying “large studies”, could you please indicate how many patients were involved in each study.
Line 36-37: You have not told the reader what ‘the presumed sources’ for endophthalmitis are for humans, please add this to the introduction.
General: It would make the introduction stronger if you added a section about topical antibiotic. You do not mention this treatment option at all and it make the inexperienced reader believe that no antibiotic is used at all (humans and veterinary) which is not true.

Materials and Methods
Line 72-73: What was the reasoning for excluding other topical antibiotics that NeoPolyBac?
Line 72-73: The use of ointment can cause phacoclastic uveitis if entering the anterior chamber (therefore, some would say that it is contraindicated to use ointment pre-and post-phaco) – has this been taken into consideration for the cases that was diagnosed with endophthalmitis.
Line 87: Eight and not 8.
Line 88: Two ACVO dipl., six residents.
Line 88-90: Which phaco machines were used and which type of pumps are they (peristaltic vs. venturi) throughout the study?
Line 92: Seven and not 7 … this a general recommendation for all numbers below ten, throughout the manuscript!
Line 93-94: I am confused – were you including NeoPolyGram or NeoPolyBac?? (see line 72-73).
Line 105-119: This is a section that should be added to the result section! Materials and Methods should also reveal WHAT cases you are including (or excluding), not how many (n), or which antibiotics you found in their records.
Line 126: You should add that eyes that was diagnosed with endophthalmitis was enucleated (if they were), and then start talking about histopathology on these enucleated eyes.
Line 128-133: You also need to add your descriptive data like mean and range for some of your data such as follow-up time, age etc.
General: I would like the inclusion criteria for a diagnosis of ‘endophthalmitis’ being explained better – clinical findings, culture, cytology, histopathology (due to enucleated) … be very specific since this is the purpose of your study!!
What about urinalysis and culture – especially on your diabetic cases! This goes back to your introduction where you said that the causes for endophthalmitis is the same for dogs as for humans, but you never told the causes for endophthalmitis for humans!

Results
Line 139: Table ?? Please give table a number (even if it is the only table you have).
Line 143-151: It is important to add diabetic versus non-diabetic dogs into this section since diabetic dogs have a higher risk for infection generally.
Line 163: Triple antibiotic solution = NeoPolyGram or NeoPolyBac … you can’t just use a new word for antibiotics – please be consistent throughout your manuscript.
Line 184: Aqueous humor aspirate = aqueous paracentesis.
Line 194-220. I am not convinced this case had endophthalmitis!!! It is going to be very hard to convince me since you do not have an aqueous culture result – I would take it out of the positive endophthalmitis group unless you can convince me WHY you think this dog had endophthalmitis!!
Line 221-248: Still not an aqueous culture result to back up your suspicion of endophthalmitis – the clinical findings are more consistent with endophthalmitis than your Case 2 but it is still a weak point that no culture is present!

Discussion
Line 263: Try not to use we and our in a scientific manuscript!
Line 270: What does this mean? (i.e., POD 2-7)
Line 288-289: An explanation is needed for each underlaying cause = Choice of surgical technique, corneal incision and intraocular lens type have also been shown to influence the rate of post-cataract surgery endophthalmitis in humans.
Line 290-291: This is not a correct way for a ref: ESCRS 291 Endophthalmitis Study Group, 2007.
Line 317-318: Use another word than ‘culture proven’.
Line 324-326: You need to go a little deeper into your discussion regarding septic endophthalmitis versus TASS.
Line 328: What limitations are you talking about when you are saying ‘Despite the study limitations …’. That is not a good way of starting a conclusion since it make the reader not believe what you have been documenting in your manuscript!

---

## Round 0.2 · accepted · Accept

I consider that the comments of reviewer 1 were appropriately addressed, and reviewer 2 has indicated that they are satisfied that your paper can be published in the current form.

Reviewer 2 ·

Basic reporting

No comments

Experimental design

No comments

Validity of the findings

No comments

Additional comments

Very nice written manuscript that gives important information to the veterinary ophthalmology profession. You have addressed my concerns well and I have no other comments - except 'good job'.